# Systemic Review of Clot Retraction Modulators

**DOI:** 10.3390/ijms241310602

**Published:** 2023-06-25

**Authors:** Alaina Guilbeau, Rinku Majumder

**Affiliations:** 1LSUHSC School of Medicine, Public University, New Orleans, LA 70112, USA; agui15@lsuhsc.edu; 2Department of Interdisciplinary Oncology, New Orleans, LA 70112, USA

**Keywords:** clot retraction, platelet activation, retraction modulators, thrombosis, actin remodeling, Protein S

## Abstract

Through a process termed *clot retraction*, platelets cause thrombi to shrink and become more stable. After platelets are activated via inside-out signaling, glycoprotein αIIbβIII binds to fibrinogen and initiates a cascade of intracellular signaling that ends in actin remodeling, which causes the platelet to change its shape. Clot retraction is also important for wound healing. Although the detailed molecular biology of clot retraction is only partially understood, various substances and physiological conditions modulate clot retraction. In this review, we describe some of the current literature pertaining to clot retraction modulators. In addition, we discuss compounds from *Cudrania trucuspidata*, *Arctium lappa*, and *Panax ginseng* that diminish clot retraction and have numerous other health benefits. Caffeic acid and diindolylmethane, both common in plants and vegetables, likewise reduce clot retraction, as do all-trans retinoic acid (a vitamin A derivative), two MAP4K inhibitors, and the chemotherapeutic drug Dasatinib. Conversely, the endogenous anticoagulant Protein S (PS) and the matricellular protein secreted modular calcium-binding protein 1 (SMOC1) both enhance clot retraction. Most studies aiming to identify mechanisms of clot retraction modulators have focused on the increased phosphorylation of vasodilator-stimulated phosphoprotein and inositol 1,4,5-triphosphate receptor I and the decreased phosphorylation of various phospholipases (e.g., phospholipase A2 (PLA_2_) and phosphatidylinositol-specific phospholipase Cγ2 (PLCγ_2_), c-Jun N-terminal kinase, and (PI3Ks). One study focused on the decreased phosphorylation of Sarcoma Family Kinases (SFK), and others have focused on increased cAMP levels and the downregulation of inflammatory markers such as thromboxanes, including thromboxane A2 (TXA_2_) and thromboxane B2 (TXB_2_); prostaglandin A2 (PGE2); reactive oxygen species (ROS); and cyclooxygenase (COX) enzyme activity. Additionally, pregnancy, fibrinolysis, and the autoimmune condition systemic lupus erythematosus all seem to affect, or at least have some relation with, clot retraction. All the clot retraction modulators need in-depth study to explain these effects.

## 1. Introduction

Clot retraction is vital in wound healing and scar formation [1]. Platelet activation and retraction not only stabilize the clot and facilitate organization of the thrombus, but activation also causes the release of growth factors, such as transforming growth factor-beta 1 (TGF-β1), epidermal growth factor (EGF) and vascular endothelial growth factor (VEGF), matrix metalloproteases 1, 2, 3, 9, and 14, ADAMTS 13, and various elastases. Singh et al. showed that interleukin (IL)-6 released by platelets induced semaphorin 7A (SEMA7A), angiopoietin-like 4 (ANGPLT4), IL-32, and matrix metalloproteinase-2 (MMP-2) expression in nearby keratinocytes in vitro [2]. Additionally, Samson et al. suggested that the contractile nature of platelets may help to re-establish the patency of small vessels after thromboembolic events and plaque rupture [3]. Clot retraction is an important platelet function in the fine balance between hemostasis and hemorrhage. The detailed mechanism of clot retraction has become a topic of highly active research, yet there is still much to learn about the associated signaling pathways and the ways in which the process can be modified. The goal of this literature review is to describe some of the compounds and conditions that affect the rate and intensity of clot retraction.

Platelets express G-protein-coupled receptors that bind thrombin, TXA2, ADP, epinephrine, and collagen to activate inside-out signaling [4]. The G-protein activates phospholipase α, β, or γ, which causes a rise in diacylglycerol and inositol 1,4,5-triphosphate. These two molecules activate protein kinase C, which, in turn, activates guanine nucleotide exchange factor. Then, GTPase Ras-related protein 1 (Rap 1) binds to an adaptor molecule, which activates the cytoskeletal proteins talin and kindlin. The grand finale of inside-out signaling is a conformational change in the highly expressed glycoprotein αIIbβIII (one study found up to 100,000 on each platelet) [4], which then acquires a greater affinity for fibrin and fibrinogen. This process is shown against the grey background in Figure 1.

Glycoprotein αIIbβIII binding to fibrinogen initiates the outside-in signaling cascade that leads to a change in actin architecture. Calpain cleaves the non-receptor kinase, src proto-oncogene known as non-receptor tyrosine kinase (Src) from the integrin β3 tail of GP-αIIbβIII, which activates multiple signaling cascades that involve the Rho-family small GTPases, phosphatidylinositol 3-kinase, focal adhesion kinase, Syk kinase, several phosphatases, and, notably, talin and kindlin, the same proteins involved in inside-out signaling [5]. Talin, kindlin, tensin, and vinculin form a complex with Src and connect to actin, causing dynamic changes in the platelet cytoskeleton, as shown against the white background in Figure 1. In this way, the platelet transforms from a flat shape into a sphere. When this transformation occurs with many platelets at once, it dramatically reduces the size of the clot.

## 2. Methods

We conducted a literature search of Pubmed and ResearchGate using the keywords “clot retraction,” “clot contraction,” “platelet contraction,” “platelet actin,” “thrombus size,” and “platelet activation.” In total, 1621 works were identified from PubMed, and 3 were identified from ResearchGate. Only peer-reviewed, full-text articles published in the previous six years (2018–2023) was used for this review; thus, 1442 items were excluded in the screening process using automated tools in the search engines. A total of 182 full-text articles were assessed for eligibility. Of these, 145 were excluded because they did not describe modulators of clot retraction (out of scope), and 3 were excluded due to insufficient rigor of evidence. The 24 articles identified as products of this selection process were used in this review article. This selection process is shown in a flow diagram in Figure 2.

## 3. Results and Discussion

### 3.1. Exogenous Clot Retraction Modulators

Chinese Mulberry *Cudrania trucuspidata* is a plant that has been used in East Asian medicine for centuries. Some of its powers include reducing inflammation [6,7] and anti-cancer [8,9] and anti-obesity effects [10,11,12]. It has traditionally been used to treat mumps, eczema, tuberculosis, and arthritis. Yet, its physiological effects do not stop there. Recent studies have provided evidence showing that several compounds in *C. tricuspidata* are capable of reducing clot retraction.

Shin et al. found that derrone, an isoflavone in *C. tricuspidata*, inhibited clot retraction through the phosphorylation of platelet vasodilator-stimulated phosphoprotein (VASP) and inositol 1,4,5-triphosphate receptor I (IP_3_RI) and the dephosphorylation of cytosolic phospholipase A_2_, p38, c-Jun N-terminal kinase, and (PI3Ks) [13]. Other downstream effects included decreases in calcium mobilization, platelet aggregation, and glycoprotein αIIbβIII affinity. Additionally, the release of TXA2 and serotonin was downregulated in the presence of derrone, whereas cAMP and cGMP were upregulated. Shin et al. showed that Cudraxanthone B, a *C. tricuspidata* xanthone, inhibited clot retraction in a manner seemingly identical to derrone, and it had the same effect on platelet signaling pathways [14]. In Table 1, one can see an organized scheme of the proposed mechanisms behind each exogenous modulator.

Shin et al. discovered yet another substance in *C. tricuspidata*, a flavone called artocarpesin, that inhibited clot retraction. Artocarpesin produced all the same effects on signaling as derrone and Cudraxanthone B [15]. The studies were all performed in vitro using washed platelets. Clot retraction was measured via sequential digital imaging and computer analysis, platelet aggregation was measured as percent light transmission, calcium mobilization was measured with a spectrofluorometer, signaling molecules were measured via immunoblotting, and serotonin and TXA2 release was measured via ELISA or EIA. In each study, the investigators analyzed cell death by measuring extracellular lactate dehydrogenase; they did not find any significant cytotoxicity. Notably, VASP and agonist-evoked inositol trisphosphate receptor (IP3-RI) phosphorylation were only affected at high concentrations (>30 μM) of cudraxanthone B and derrone, and artocarpesin only affected IP3-RI phosphorylation at concentrations of 80–100 μM.

Nam et al. identified morin hydrate as another clot retraction inhibitor from *C. tricuspidata* [16]. Morin hydrate can also be isolated from white mulberry *Morus alba* and the almond tree *Prunus dulcis*. The platelet-signaling effects of morin hydrate are similar to the effects of the aforementioned compounds, with the addition of the downregulation of P-selectin and ATP, as well as the decreased phosphorylation of (PLCγ2) and extracellular signal-regulated kinase (ERK). Effects on the phosphorylation of VASP were not described for morin hydrate. Interestingly, Nam et al. also measured the effect of morin hydrate on blood coagulation using activated partial prothrombin time (aPTT) and prothrombin time (PT) assays; the investigators did not find any significant difference compared with the control. This result suggested that morin hydrate is a potential therapeutic agent for reducing the risk of plaque rupture and harmful platelet accumulation in vessels, without the risk of hemorrhage associated with the current anticoagulant therapies. Further studies should be conducted to evaluate the efficacy of *C. tricuspidata* compounds in mice or other animal models for the prevention of thrombotic events such as stroke and myocardial infarction. *Cudrania trucuspidata* is a valuable plant that could be used to benefit countless patients who have cardiovascular disease; its use should be integrated into Western medicine just as it has been integrated into Asian medicine for many years.

Kwon et al. investigated arctigenin, a ligand that inhibits clot retraction [17]. Arctigenin is found in *Arctium lappa*, commonly called greater burdock, of the family Asteraceae. Arctigenin reduced thrombin-, ADP-, and collagen-induced platelet aggregation, ATP and serotonin release, calcium mobilization, fibrinogen binding to GP-αIIbβIII, P-selectin expression, TXA_2_ production, and COX-1 activity. It also increased the phosphorylation of VASP and IP3-RI and the production of cAMP. There were no significant effects on coagulationm as measured with aPTT and PT assays. A lactate dehydrogenase assay again did not show any cytotoxicity of arctigenin. Like *C. tricuspidata*, *A. lappa* could be a therapeutic agent for the prevention of atherosclerosis. Arctigenin has the additional benefits of being an anti-cancer agent [18,19,20]; it promotes renal health [21,22], and it possibly thwarts depression by inhibiting glial cells [23].

Caffeic acid is a polyphenol that occurs in many plants and has been studied in mice because of its anti-thrombotic effects. Nam et al. reported that caffeic acid appeared to be anti-platelet in nature, which included the inhibition of clot retraction [24]. The investigators focused on increased cAMP production in the presence of caffeic acid. Dipyridamole, a phosphodiesterase 3 inhibitor, also reduced clot retraction in vitro. Thus, Nam et al. concluded that the major action of caffeic acid was increasing cAMP levels in platelets, which led to the downstream phosphorylation of signaling molecules such as VASP and IP3-RI. This conclusion would explain how so many different substances can affect a multi-faceted process such as clot retraction in a seemingly identical manner.

Indole-3-carbinol is a phytochemical in cruciferous vegetables such as cabbage, kale, broccoli, and Brussel sprouts. It is sometimes taken as a supplement because of its anti-inflammatory [25,26], anti-cancer [27,28,29], and anti-thrombotic [30,31] effects. However, Ramakrishna et al. focused on diindolylmethane, a metabolite of indole-3-carbinol that has more potent effects on platelets and clot retraction compared with its parent compound [32]. Diindolylmethane was first analyzed via computer simulation. It was discovered that glycoprotein VI and purinergic receptor Y12 on platelets were modified by both indole-3-carbinol and diindolylmethane, but diindolylmethane had a greater effect. Clot retraction and platelet aggregation were both reduced substantially by diindolylmethane in vitro.

In the in vivo studies, FeCl_3_-induced injury of the carotid artery occluded more slowly in mice pre-treated with diindolylmethane, and platelets from these mice produced lower levels of ROS, TXB_2_, COX-1, and PGE_2_ and higher levels of cAMP. Vessel occlusion was measured via ultrasound. Because the study was performed in vivo, clot retraction was measured based on the weight and size of the thrombus, collected from the carotid artery instead of using the conventional sequential imaging technique. Weight and size measurements are not the most accurate approach, because these measurements do not indicate the difference between the initial size of the clot and the size after retraction. Multiple time points are needed to properly conclude that clot retraction is inhibited in vivo. However, the study did show that diindolylmethane had a major anti-platelet effect.

*Panax ginseng* is another herb used in traditional East Asian medicine. Kwon et al. found that Ginsenoside Ro, a compound in this plant, inhibited thrombin-induced platelet aggregation, clot retraction, and fibronectin adhesion to GP- αIIbβIII [33]. This study focused on the Akt and PI3K pathways, which were proposed to be the sites of action of Ginsenoside Ro. The Akt phosphorylation inhibitor miltefosine [34], which is commonly used to treat leishmaniasis, in addition to the PI3K phosphorylation inhibitor wortmannin [35], were used as positive controls. Miltefosine and wortmannin decreased platelet aggregation, clot retraction, and fibronectin binding, and in combination with Ginsenoside Ro, they had synergistic effects on each process. Ginsenoside Ro, alone, inhibited Akt and PI3K phosphorylation, as assessed via immunoblotting. This study was limited because fibrinogen binding to GP-αIIbβIII was measured at only one concentration of Ginsenoside Ro (300 μM), whereas fibronectin adhesion and other parameters were measured at multiple concentrations. Similarly, the anti-platelet effects of miltefosine and wortmannin were only shown at one concentration (10 μM). More reliable conclusions could have been drawn if these effects had been shown to be dose-dependent. Although the study conducted by Kwon is an important first step in assessing the activity of Ginsenoside Ro, the speculation that it mainly affects the Akt and PI3K pathways should be expanded to include pathways implicated by other clot retraction inhibitors—pathways such as the upregulation of cAMP, downregulation of COX1, TXA2/TXB2, and ROS, and the increased phosphorylation of VASP and IP_3_RI.

Luo et al. proposed that all-trans retinoic acid (ATRA, also known as tretinoin) reduced clot retraction by causing diminished Syk and PLCγ2 phosphorylation [36]. This effect is significant because the previously discussed morin hydrate’s proposed mechanism also included diminished PLCγ2 phosphorylation; possibly, these effects are clues that PLCγ2 has a central function in the regulation of clot retraction. This idea is further supported by PLCγ2′s upstream position in the process of inside-out signaling [5,37]. Notably, the investigators found ATRA to be a hemophilic agent, in contrast to morin hydrate, which did not affect coagulation time, although the effect of ATRA on coagulation was measured based on mouse tail bleeding time, while the effect of morin hydrate was measured via coagulation assays.

An interesting study conducted by Debreceni et al. showed that the chemotherapeutic drug Dasatinib reduces clot retraction by attenuating Sarcoma Family Kinase (SFK) phosphorylation [38]. Dasatinib is a variation of Imatinib, the targeted therapy for Chronic Myelogenous Leukemia (CML). It works by inhibiting the BCR-ABL tyrosine kinase, an aberrant fusion protein expressed in CML. Hemophilic adverse effects have been documented in many patients taking Dasatinib, which is thought to be due to platelet dysfunction, as well as pancytopenia [39]. This study focused on the molecular mechanism involved in platelet dysfunction. When added to healthy patient plasma, Dasatinib was shown to decrease phosphatidyl serine exposure in response to convulxin, total thrombin generation, and peak thrombin and to increase lag time and the time required to reach peak thrombin. Platelets treated with Dasatinib also had lower convulxin-induced GP-αIIbβIII expression (measured via flow cytometry) and clot retraction (measured via the percent of extruded serum). This research was different from the previously discussed studies, since the effects were shown in the coagulation cascade, along with platelets. Western blotting was then performed to demonstrate that Dasatinib-treated plasma had lower levels of SFK phosphorylation at the C-terminal tail and the active loop. The attenuation of SFK phosphorylation was demonstrated in Src, Fyn, and Lyn. These results enhance our understanding of clot retraction by solidifying the central role of SFKs in outside-in signaling.

Lastly, Nam et al. found that the mitogen-activated protein kinase 4 (MAP4K4) inhibitors GNE 495 and PF06260933 decreased clot retraction, with the same downstream effects as those observed with many of the aforesaid compounds. These effects included lower TXB2, ATP, serotonin, and fibrinogen binding and higher cAMP, IP3-RI, and VASP phosphorylation [40]. Protein kinase A catalytic subunit (PKAc) phosphorylation was an additional parameter measured in this study; PKAc was increased by both MAP4K4 inhibitors. There was no significant effect on total COX activity, clotting time (aPTT and PT), or cytotoxicity. As in the caffeic acid study, dipyridamole was used as a positive control, and it was found to work synergistically with GNE 495, providing further evidence that cAMP is a major component of clot retraction regulation.

### 3.2. Endogenous Modulators

In a cross-sectional study of 100 pregnant and 100 non-pregnant women, Okoroiwu et al. observed that both the clot retraction time and platelet counts were lower in the pregnant women [41]. The authors concluded that pregnancy and, possibly, estrogen and progesterone levels influence clot retraction. However, more evidence is necessary to support this conclusion. In this study, clot retraction time was measured only at 1, 2, 4, and 24 h intervals, and the time recorded for each sample was “the length of time it took for the clotted blood to retract,” with no parameters for what it entailed. Altered platelet activation during pregnancy is an interesting phenomenon of uncertain significance, and it should be further investigated using conventional clot retraction assays, such as volume measurement at short intervals (10–20 min) with photography or force transduction technology [4].

Fibrinolysis is another physiological condition proposed to have a relationship with clot retraction. In an in vivo study, Andre et al. found that fibrinolysis markedly increased clot retraction [4]. Mice mesenteric veins were injured using a needle, and then thrombin was injected into the circulation to promote clot formation. Clot retraction was measured via confocal fluorescence microscopy. When fibrinolysis was initiated through the addition of a small amount of tissue plasminogen activator (≤500 pM), clot retraction was enhanced. Additionally, clot retraction was measured in the presence of six fibrinolysis inhibitors: tranexamic acid, thrombin-activated fibrinolysis inhibitor, α2 antiplasmin, Factor XIIIa inhibitor, and two different IgG antibodies acting as tPA antagonists. Each of these fibrinolysis inhibitors were found to individually reduce clot retraction.

Curiously, Tutwiler et al. reported the exact opposite findings, i.e., clot retraction influenced the rate of fibrinolysis [42]. The authors inhibited clot retraction with either blebbistatin, a myosin IIa inhibitor, abcixumab; an αIIbβIII inhibitor; or latrunculin A, which prevents actin polymerization. The result was a 4-fold decrease in external fibrinolysis and a 2-fold increase in internal fibrinolysis. Obviously, fibrinolysis and clot retraction are processes that interact in some way, perhaps even in a bidirectional manner, but the exact mechanism is currently unclear.

As demonstrated for many of the exogenous modulators described earlier, inflammatory markers such as PGE2 and TXB2/TXA2 are closely linked to clot retraction. Thus, it is not surprising that Misztal et al. found a dose-dependent (50–500 μM) reduction in clot retraction mediated by the myeloperoxidase product hypochlorite [43]. At higher concentrations (125–500 μM), hypochlorite reduced fibrinolysis. Platelet aggregation, thrombus formation (in artificial flow chambers that mimicked blood vessels), ATP production, and P-selectin expression were also diminished in the presence of hypochlorite. Interestingly, hypochlorite affected the structure of the clot, as assessed via confocal microscopy, and when fibrinogen was added to hypochlorite solution, di-tyrosine crosslinks were observed as a possible mechanism for the aforesaid phenomena. This could have important implications for patients who have chronic inflammatory states such as tuberculosis, diabetes, and autoimmune diseases.

Regarding autoimmune diseases, Le Minh et al. observed that clot retraction was lowered in patients with systemic lupus erythematosus (SLE), but not because of hypochlorite [44]. The authors compared 51 lupus and 60 healthy patient blood samples, and they found lower clot retraction and lower P-selectin expression and fibrinogen–GP-αIIbβIII binding capacity in the SLE patients. In fact, patients with higher levels of anti-dsDNA antibodies had the greatest reduction in clot retraction. This finding prompted an in vitro experiment wherein normal blood was incubated with autoantibodies from patients with lupus prior to clotting; the platelets seemed to be continuously stimulated by the antibodies (increased clot retraction), and then the platelets rapidly lost activity (decreased clot retraction). The authors suggested that the lupus autoantibodies bind to and “wear out” platelets, which suggests the dysregulation of clot retraction in SLE patients.

On the topic of inflammatory conditions, another study investigated a certain matrix protein called secreted modular calcium-binding protein 1 (SMOC1), which is overexpressed in platelets from individuals with type II diabetes mellitus [45]. Since SMOC1 is regulated by microRNA-223, which is abundantly expressed in platelets, it was theorized that SMOC1 might have some function in platelet activation and activity. Thus, Lagos et al. used mice platelets with one SMOC1 allele knocked out (SMOC1^+/−^) to measure clot retraction, platelet aggregation, platelet spreading, leukocyte–platelet aggregations, and thrombin-induced calcium mobilization. All these parameters were significantly lower in the SMOC1^+/−^ platelets compared to the wild type, implicating that this protein may play a role in the pro-thrombotic state of diabetic patients. Platelet hyperactivity was further evidenced by increased β1 integrin phosphorylation, as well as aggregation in platelets isolated from diabetic patients, and these effects were mitigated with the addition of anti-SMOC1 antibody. This is an invaluable finding, since we know that myocardial infarction is the leading cause of death in diabetic individuals, and these people have a higher risk of having such an event than the non-diabetic population [46]. Further studies should solidify this correlation using a larger number of diabetic blood samples, as this study only included 16.

One last significant finding pertains to the endogenous anticoagulant Protein S. Protein S is mostly synthesized by hepatocytes, but it is also expressed in platelets at a low level. Genetically modified Cre-Lox mice were created using the Platelet Factor 4 promotor to remove the *PROS1* gene (Protein S gene) in platelets while preserving it in other tissues. The purpose of this study, conducted by Calzavarini et al. [47], was to show that the inactivation of platelet Protein S expression leads to increased thrombus formation in the vena cava. In a low thrombin condition, clot retraction was not affected, but with high concentrations of thrombin (10 U/mL), clot retraction was impaired in the Pros1^lox/lox^Pf4-Cre^+^ mice. In another study, Brouns et al. [48] used real-time multi-color microscopic imaging to compare 23 Protein-S- or Protein-C-deficient patient blood samples with samples from 15 healthy persons. The former group exhibited less platelet activation, fibrin formation, and clot retraction, which indicated that platelet Protein S may enhance clot retraction in humans. In vitro assays with normal blood and the addition of Protein S or anti-PS antibody could be the next step in confirming this theory.

### 3.3. Discussion

Clot retraction is an interesting and medically relevant topic, and understanding its regulation is especially important, since cardiovascular disease is the leading cause of death worldwide [49]. We described some recent research concerning clot retraction modulation and physiological conditions that can affect this process. Notably, *C. trucuspidata*, with four different compounds that independently inhibit clot retraction, is a potential therapeutic agent for reducing atherosclerosis and plaque rupture. Other plant substances such as arctigenin, caffeic acid, diindolylmethane, and Ginsenoside Ro have potential as therapeutic agents. The CML treatment Dasatinib seems to inhibit platelet function, which can cause adverse events such as bleeding. Pregnancy, fibrinolysis, systemic lupus erythematosus, and the ROS hypochlorite all seem to influence clot retraction, although some of these mechanisms are clearer than others, and the overexpression of the SMOC1 matrix protein seems to contribute to abnormal platelet activity in diabetic patients. Additionally, there is evidence that platelet Protein S facilitates clot retraction in mice and humans.

## 4. Conclusions

Although we do not know all the pathways involved in clot retraction, it is promising to see such a diverse collection of pertinent research in the literature. Indeed, more biologists, healthcare workers, pharmaceutical companies, and researchers should know about the kinetic nature of platelets, as they are necessary for a complete understanding of thrombosis and hemodynamics. Clot retraction is vital for the stabilization of the thrombus, but clot retraction also maintains openings in vessels after coagulation and initiates wound healing after trauma. We expect that this review will encourage the sharing of ideas and increase interest in the topic of clot retraction.

## Figures and Tables

**Figure 1 ijms-24-10602-f001:**
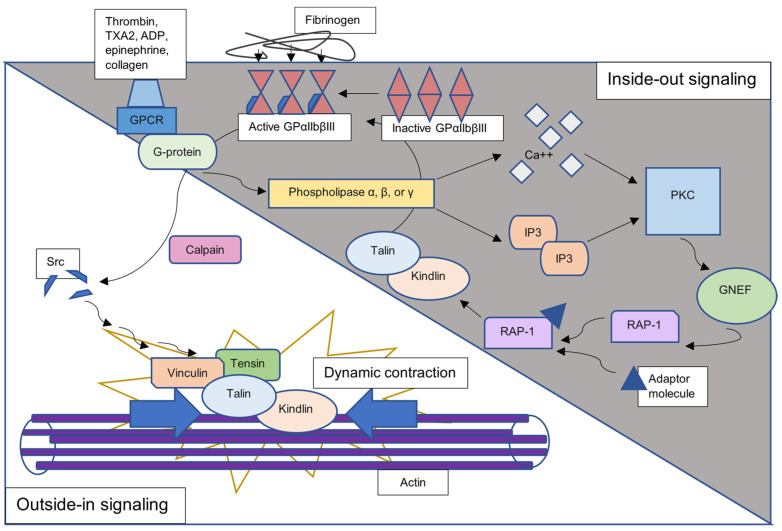
Clot retraction signaling cascade. G-protein-coupled receptor binds thrombin, TXA2, ADP, epinephrine, and collagen, activating various phospholipases. Intracellular calcium and IP3 activate protein kinase C, which signals RAP-1 to bind an adaptor molecule. Talin and kindlin produce a conformational change in GP-αIIbβIII, increasing its affinity for fibrinogen. This process of inside-out signaling is shown against the grey background. Src is cleaved from the β3 tail of GP-αIIbβIII and, through various signaling pathways (not shown), forms a complex with tensin, vinculin, talin, and kindlin. Actin rearrangement occurs, which ultimately causes platelet contraction and stabilizes the clot. Outside-in signaling is shown against the white background.

**Figure 2 ijms-24-10602-f002:**
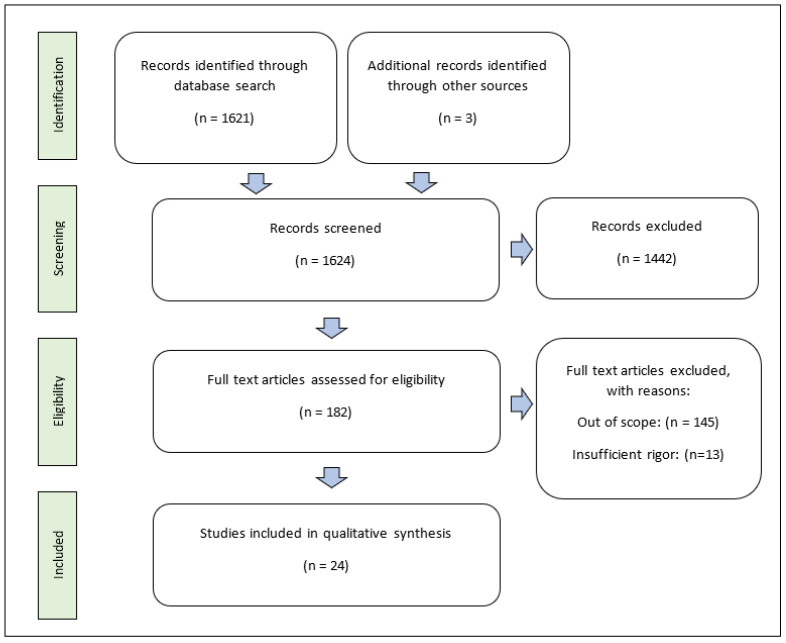
PRISMA diagram of literature qualification process.

**Table 1 ijms-24-10602-t001:** Proposed downstream effects of exogenous modulators.

Origin	Compound	Increased	Decreased
*Cudrania tricuspidata*	derrone	p-VASP, p-IP_3_RI, cAMP, cGMP	p-cPLA_2_, p-p38, p-JNK, p-PI3K, Ca^++^, Plt_Agg_, αIIbβIII_aff_, TXA_2_, 5-HT
	Cudraxanthone B		
	artocarpesin		
	morin hydrate	p-IP_3_RI, cAMP, cGMP	P-selectin_ex_, ATP, p-PLCγ2, p-ERK, p-cPLA_2_, p-p38, p-JNK, p-I3K, Ca^++^, Plt_Agg_, αIIbβIII_aff_, TXA2, 5-HT
*Arctium lappa*	arctigenin	p-VASP, p-IP_3_RI, cAMP	Plt_Agg_, ATP, 5-HT, Ca^++^, αIIbβIII_aff_, P-selectin_ex_, TXA_2_, COX-1_act_
Many plants	caffeic acid	cAMP, p-VASP, p-IP_3_RI	
Cruciferous vegetables	diindolylmethane	GPVI_mod_, PRY12_mod_, cAMP	ROS, TXB_2_, COX-1, PGE_2_
*Panax ginseng*	Ginsenoside Ro	Plt_Agg_, αIIbβIII_aff_,	p-Akt, p-PI3K,
Synthetic	all-trans retinoic acid		P- PLCγ2, p-syk
Synthetic	Dasatinib		p-Src, p-Fyn, p-Lyn, PS_ex_, thrombin, αIIbβIII_ex_
Synthetic	GNE 495 and PF06260933	p-VASP, p-IP_3_RI, cAMP, p-PKAc	TXB_2_, ATP, 5-HT, αIIbβIII_aff_,

The abbreviation p- indicates phosphorylation. Plt_Agg_ describes platelet aggregation. Ca^++^ indicates calcium mobilization from intracellular stores. αIIbβIII_aff_ describes GP-αIIbβIII affinity for fibrinogen or fibronectin binding, while αIIbβIII_ex_ describes the level of expression. Serotonin is abbreviated as 5-HT. P-selectin_ex_ describes P-selectin expression. COX-1_act_ refers to COX-1 activity. GPVI_mod_ and PRY12_mod_ indicate glycoprotein VI and purinergic receptor Y12 modification, respectively (unspecified changes). PS_ex_ describes phosphatidyl serine expression.

## Data Availability

This is a review based on the published literature.

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
