# Peer review of "Systemic Review of Clot Retraction Modulators"

_ijms, 2023, doi:10.3390/ijms241310602_

Round 1
Reviewer 1 Report
In this review, ,, Clot Retraction Modulators,, by Alaina Guilbeau et al. , the authors described some of the current literature pertaining to clot retraction modulators.
Comments and Suggestions for the Authors
The article has some shortcomings:
- I made some suggested changes in the body of the manuscript;
- Authors should double-check abbreviations: Abbreviations are explained when they first appear in the abstract or main text and contribute to making the text easier to read and the information conveyed more efficiently. Once an abbreviation has been established and explained, it will be used throughout the entire manuscript, with the exception of the abstract, where it must be treated separately. Please revise the whole manuscript and explain the abbreviations used directly, without explanation;
- Authors must provide in the Materials and Methods section:
· the Data Sources and Search Strategy (The electronic databases PubMed, EMBASE, and Cochrane Library..............)
· Study Selection and Eligibility Criteria,
· Data Extraction (We extracted the following data from the included studies: author’s name, publication year, country, total subjects, study population, ..........)
- Authors must provide in the Results section, a Subsection - Literature Search with Figure 2 outlines the PRISMA flow diagram describing the identification, screening, and inclusion phases.
- The design of the research project must be improved.
- The methods must be improved.
- The results are presented clearly but can be improved.
- Discussions must be substantially improved.
- Authors must provide a Conclusions section
Overall Recommendation: Accept after major revision.

Minor editing of English language required
Author Response
We appreciate the valuable suggesstions from reviewer 1. The manuscript has been improved after incorportaing changes suggessted by Reviewer 1.Here are the point to point response to Reviewer 1.
Point 1
Authors should double-check abbreviations: Abbreviations are explained when they first appear in the abstract or main text and contribute to making the text easier to read and the information conveyed more efficiently. Once an abbreviation has been established and explained, it will be used throughout the entire manuscript, with the exception of the abstract, where it must be treated separately. Please revise the whole manuscript and explain the abbreviations used directly, without explanation;
Response: We have revised the whole manuscript and followed reviewers suggesstions.
Point 2
Authors must provide in the Materials and Methods section:
- the Data Sources and Search Strategy (The electronic databases PubMed, EMBASE, and Cochrane Library..............)
- Study Selection and Eligibility Criteria,
- Data Extraction (We extracted the following data from the included studies: author’s name, publication year, country, total subjects, study population, ..........)
Reponse: We have incorporated all the suggesstions mentioned by this reviewer in Materials Methods Section. We added a new Figure, Figure 2- PRISMA diagram of literature qualification process in the Materials Methods secion.
Point 3
The design of the research project must be improved.
- The methods must be improved.
- The results are presented clearly but can be improved.
- Discussions must be substantially improved.
Response: Changes have been made throughout the manuscript to incorporate changes suggested by Reviewer 1.
Point 4
Authors must provide a Conclusions section
Response: We added a Conclusion section in the revised manuscript.
Reviewer 2 Report
This is a good manuscript on technically correct and sums up interesting information.
From my point of view, this article should be developed, by including a larger number of summary tables regarding the topic addressed (the only table in the current manuscript is not written according to the rigors of the journal). Also, the Figure should be improved for better visibility. I would recommend improving the number of cited articles, for a review it is not enough to cite only 49 bibliographic sources. The subject is interesting and deserves to be developed, the work would bring benefits by being present in the scientific flow, but it needs to be developed, structured and better illustrated.
Author Response
We appecitae the valuable suggesstions from the Reviewer 2.
Here are the point to point response to the comments from Reviewer 1.
Point 1
This article should be developed, by including a larger number of summary tables regarding the topic addressed (the only table in the current manuscript is not written according to the rigors of the journal).
Response: We reformatted the table and added a new Figure to improve the manuscript.
Point 2
Also, the Figure should be improved for better visibility.
Response: We have improved the Figure 1 by changing the resolution to 600dpi.
Point 3
I would recommend improving the number of cited articles, for a review it is not enough to cite only 49 bibliographic sources.
Response: We have added 4 new references.
Round 2
Reviewer 1 Report
Please double-check the abbreviations and make the necessary corrections so that the abbreviations are explained when they first appear, both in the abstract and in the manuscript text, table and figure legends.
In the legend of the table, all the abbreviations used in the table must be explained, not just a part of them.

Minor editing of English language required
Author Response
The answers to reviewer's 1 comments are listed here
P1- Please double-check the abbreviations and make the necessary corrections so that the abbreviations are explained when they first appear, both in the abstract and in the manuscript text, table and figure legends
Answer- We have made sure that the abbreviations are explained when they first appear, both in the abstract and in the manuscript text, table and figure legends.
P2-In the legend of the table, all the abbreviations used in the table must be explained, not just a part of them.
Answer- We have now explained all the abbreviations in the legend of the table.
P3- Minor editing of English language required.
Answer- The manuscript has been extensively revised by a professional editor, Dr. Howard Fried who owns , Cursor Scientific Editing and Writing, L. L. C. The changes are kept in the manuscript.